# The chemokine CXCL13 in lung cancers associated with environmental polycyclic aromatic hydrocarbons pollution

Gui-Zhen Wang[1†], Xin Cheng[1†], Bo Zhou[1], Zhe-Sheng Wen[2], Yun-Chao Huang[3], Hao-Bin Chen[4], Gao-Feng Li[3], Zhi-Liang Huang[2], Yong-Chun Zhou[3], Lin Feng[5], Ming-Ming Wei[1], Li-Wei Qu[1], Yi Cao[6], Guang-Biao Zhou[1*]

[1]State Key Laboratory of Membrane Biology, Institute of Zoology, Chinese Academy of Sciences, Beijing, China; [2]Department of Thoracic Surgery, Sun Yat-Sen University Cancer Center, Guangzhou, China; [3]Department of Thoracic Surgery, The Third Affiliated Hospital of Kunming Medical University, Kunming, China; [4]Department of Pathology, The First People's Hospital of Qu Jing City, Qu Jing, China; [5]Department of Pathology, Chinese PLA General Hospital, Beijing, China; [6]Laboratory of Molecular and Experimental Pathology, Kunming Institute of Zoology, Chinese Academy of Sciences, Kunming, China

*For correspondence: gbzhou@ioz.ac.cn

[†]These authors contributed equally to this work

Competing interests: The authors declare that no competing interests exist.

**Abstract** More than 90% of lung cancers are caused by cigarette smoke and air pollution, with polycyclic aromatic hydrocarbons (PAHs) as key carcinogens. In Xuanwei City of Yunnan Province, the lung cancer incidence is among the highest in China, attributed to smoky coal combustion-generated PAH pollution. Here, we screened for abnormal inflammatory factors in non-small cell lung cancers (NSCLCs) from Xuanwei and control regions (CR) where smoky coal was not used, and found that a chemokine CXCL13 was overexpressed in 63/70 (90%) of Xuanwei NSCLCs and 44/71 (62%) of smoker and 27/60 (45%) of non-smoker CR patients. CXCL13 overexpression was associated with the region Xuanwei and cigarette smoke. The key carcinogen benzo(a)pyrene (BaP) induced CXCL13 production in lung epithelial cells and in mice prior to development of detectable lung cancer. Deficiency in Cxcl13 or its receptor, Cxcr5, significantly attenuated BaP-induced lung cancer in mice, demonstrating CXCL13's critical role in PAH-induced lung carcinogenesis.

## Introduction

Air pollution is a diverse mixture of pollutants that originated from anthropogenic and natural sources, is comprised of particulate matter (PM), gases (e.g., sulfur oxides, carbon monoxide, ozone), organic compounds (e.g., polycyclic aromatic hydrocarbons, PAHs), metals (e.g., lead, vanadium, and nickel), and others, such as microbes (*Akimoto, 2003*; *Huang et al., 2014*). Air pollution is a global environmental health risk that affects the populations in developed and developing countries alike, and satellite observations suggest that 80% of the global population resides in locations where the ambient pollutant concentrations exceed the World Health Organization (WHO) Air Quality Guideline (*van Donkelaar et al., 2010*). Outdoor air pollution in cities and rural areas was estimated to cause 3.7 million premature deaths annually worldwide in 2012, including 220,000 deaths due to lung cancer (*WHO, 2014*). Recently, outdoor (*Loomis et al., 2013*) and indoor (*WHO, 2010*) air pollution has been classified as a Group 1 carcinogen in humans by the International Agency for Research on Cancer (IARC) of WHO. Indeed, the risk of lung cancer rises by 18% for every increase of 5 µg/m³ of PM smaller than 2.5 µm in diameter ($PM_{2.5}$) in the environment; the risk increases by

**eLife digest** Lung cancer causes the most cancer deaths worldwide. For decades, people have known that lung cancer is associated with environmental factors, and both cigarette smoke and air pollution are known to cause cancers in humans. Smoke and air pollution both contain chemicals called polycyclic aromatic hydrocarbons (or PAHs). These chemicals cause chronic inflammation of the lung, which in turn is a major risk factor for developing lung cancer.

However, it is unclear exactly how PAHs trigger inflammation and cancer. Xuanwei City in China is suited to the study of this question because until the 1970s its inhabitants used 'smoky coal' for cooking in unventilated indoor spaces; this produced high levels of small particles that contain high concentrations of PAHs. Women from this region, who traditionally do most of the cooking, have rates of lung cancer comparable to those of men. In other parts of China a woman's chance of getting lung cancer is approximately half that of a man's. Therefore, Xuanwei City provides a setting in which air pollution is a main contributor to lung cancer risk.

Wang, Cheng et al. have now compared the levels of certain proteins (which are linked to inflammation) in lung cancer patients from Xuanwei City with those in control regions of China. The level of one such protein marker, called CXCL13, was particularly high in almost all patients from Xuanwei City, but only highly expressed in half of the patients from the control regions. Moreover, there was also a clear link between cigarette smoke and CXCL13 expression because, in control regions, smokers were much more likely to have high levels of CXCL13 than non-smokers. To test whether PAHs cause CXCL13 expression, Wang, Cheng et al. first exposed normal lung epithelial cells, cancer cells and then mice to a PAH. These experiments showed that CXCL13 levels did indeed increase and the mice developed lung tumours. However, when the genes for CXCL13 or its binding partner were deleted, the mice no longer got cancer when exposed to the PAH. This shows that CXCL13 signalling is an important mechanism by which PAHs cause lung cancer. Lastly, further experiments showed that CXCL13's binding partner is highly expressed on some immune cells that can promote lung cancer.

Importantly, the over-expression of CXCL13 occurred before the lung tumours developed. This might provide a new treatment strategy in which CXCL13 signalling could be inhibited after the exposure to PAHs. Future studies may now focus on discovering new drugs, or modifying existing drugs, to achieve this goal.

22% for every increase of 10 µg/m$^3$ in PM smaller than 10 µm (PM$_{10}$) (*Raaschou-Nielsen et al., 2013*). There have been efforts to investigate how air pollution causes human cancers by exposing cells and animals to PM or related chemicals (*Straif et al., 2012*), but the carcinogenic mechanism remains elusive, at least in part, due to the difficulty of identifying human lung cancers that are causally associated with air pollution.

Xuanwei City of the Yunnan Province in China provides a significant association between air pollution and lung cancer (*Lan et al., 2002*; *Sinton et al., 1995*; *Mumford et al., 1987*). Until the 1970s, residents of this region used smoky coal in unvented indoor fire pits for domestic cooking and heating, which are processes that release high concentrations of PM$_{2.5}$/PM$_{10}$ that contain high concentration PAHs (*Mumford et al., 1987*). Indoor air pollution in Xuanwei is associated with a significant increase in the absolute lifetime risk of developing lung cancer (*Xiao et al., 2012*), and the lung cancer incidence in this region is among the highest in China (*Mumford, et al., 1987*; *Xiao et al., 2012*; *Li et al., 2011*). In particular, nearly all women in this region are non-smokers and cook food on the household stove, and the lung cancer incidence in women is high. In Xuanwei, the male-to-female ratio of lung cancer incidences is 1.09:1, while that of China's national average reaches 2.08:1 (*Xiao et al., 2012*). Tobacco smoke was weakly and non-significantly associated with lung cancer risk in this region (*Kim et al., 2014*). In the 1990s, a reduction in the lung cancer incidence was noted after stove improvement, supporting the association between indoor air pollution and lung cancer (*Lan et al., 2002*). These findings had been cited by the IARC monograph to classify indoor emissions from household coal combustion as 'carcinogenic to humans (Group 1) (*WHO, 2010*). However, lung cancer incidence in Xuanwei is still increasing (*Li et al., 2011*),

possibly due to pollutants generated by coal-burning industrial plants that have moved into the area (*Cao and Gao, 2012*). Therefore, the population in this highly polluted region (HPR) provides a unique opportunity to dissect air pollution-related lung carcinogenesis. Furthermore, the key carcinogens of smoky coal combustion are PAHs (*Lv et al., 2009*), which are also the key carcinogens of tobacco smoke that causes more than 85% of global lung cancer deaths (*Hecht, 2012*). Xuanwei lung cancer is, therefore, applicable to elucidate tobacco smoke-induced lung carcinogenesis.

Chronic inflammation can promote cancer formation, progression and metastasis by inducing oncogenic mutations, genomic instability, and enhanced angiogenesis (*Grivennikov et al., 2010*). Studies show that PAHs can cause immune suppression (*Zaccaria and McClure, 2013*) and induce the secretion of cytokines/chemokines, including tumor necrosis factor (TNF)-α, interleukin (IL)-1β, IL-6, IL-8, CC chemokine ligand 1 (CCL1), chemokine (C-X-C motif) ligand 1 (CXCL1), and CXCL5 (*N'Diaye et al., 2006*; *Umannová et al., 2011*; *Chen et al., 2012*; *Dreij et al., 2010*). These factors may facilitate cancer initiation and progression. However, most of these factors were identified in cellular and animal PAH exposure models. Clinically relevant inflammation factors should be identified to shed insights into PAH-related carcinogenesis and provide novel therapeutic targets for lung cancer.

To systematically investigate air pollution-induced lung carcinogenesis, we used Xuanwei lung cancers to analyze the abnormalities in the cancer genomes (*Yu et al., 2015*), genome-wide DNA methylation, non-coding RNAs (miRNAs [*Pan et al., 2015*] and lncRNAs), and inflammation factors. The abnormalities found in the HPR lung cancers were tested in patients from control regions (CRs) where smoky coal was not used, to compare the difference between HPR and control region (CR) lung cancers. In this study, we explored the abnormal inflammatory factors in HPR non-small cell lung cancers (NSCLCs).

## Results

### Abnormal inflammatory factors in HPR NSCLCs

Using a microarray analysis of 84 cytokines/chemokines in tumor samples and their adjacent normal lung tissues of eight HPR NSCLCs, we found that the expression of four cytokines (*IL-1F5, IL-1F9, MIF,* and *SPP1*) and seven chemokines (*CXCL13, CCL7, CCL20, CCL26, CXCL6, CXCL9,* and *CXCL14*) was increased in tumors compared with their counterpart normal lung tissues (*Figure 1A*). Among them, *CXCL13* was the most significantly up-regulated gene, with an average of 63-fold (10.48–173.65) higher expression in the tumor samples than in the normal controls. The expression of three cytokines (*IL-1α, IL-5,* and *TNF-α*) and nine chemokines (*CCL4, CCL17, CCL18, CCL21, CCL23, CXCL1, CXCL2, CXCL3,* and *CXCL5*) was lower in the tumor samples than in the normal controls (*Figure 1A*).

### Overexpression of CXCL13 in NSCLCs

We expanded these observations, and found that the expression of CXCL13 was elevated in the tumor tissues from 63/70 (90%) HPR patients (*Figure 1B*) and 71/131 (54.2%) CR patients (*Table 1*) compared with their normal controls. CXCL13 expression was much higher in the HPR patients than the CR cases (p<0.002; *Figure 1C*). Using immunohistochemistry (IHC) and immunoreactivity scoring, we showed that the expression of the CXCL13 protein was significantly higher in the tumor samples than their adjacent normal controls (*Figure 1D, E*). CXCL13 expression was higher in the HPR NSCLCs than the CR patients, and the CR smoker NSCLCs had higher CXCL13 levels than the CR non-smoker patients (*Figure 1B, D, E*, and *Table 1*). Using Western blot assays, we found that CXCL13 was much higher in the tumor samples than their adjacent normal lung tissues (*Figure 1F*). ELISA showed that the CXCL13 serum concentration was higher in the HPR patients compared with the CR patients, while the latter was higher than the healthy donors (*Figure 1G*).

### Analysis of association between CXCL13 expression and clinical characteristics

*CXCL13* expression was not significantly different in smoker and non-smoker HPR patients (p=0.17; *Table 1*), suggesting that severe air pollution had a carcinogenic effect on humans. In NSCLCs from CRs, however, the expression of *CXCL13* was significantly higher in smokers (44/71, 62%) than in

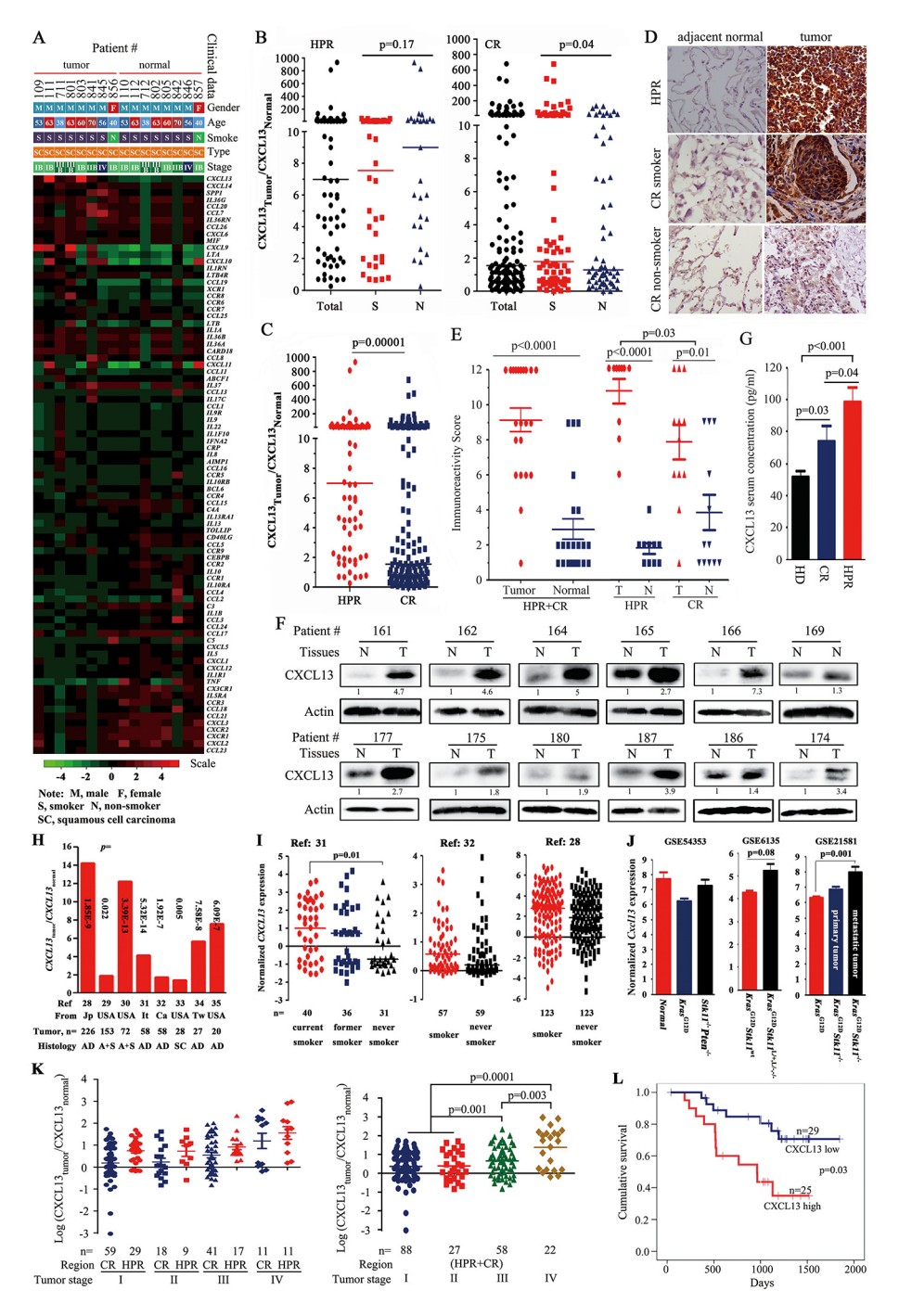

**Figure 1.** CXCL13 expression in lung cancer. (**A**) A PCR array was used to detect the expression of 84 cytokines/chemokines in eight highly polluted region (HPR) lung cancers. (**B**) The ratios of CXCL13 in tumor samples to their counterpart normal lung tissues from both the HPR and control region (CR) non-small cell lung cancers (NSCLCs). (**C**) Comparison of the $CXCL13_{tumor}/CXCL13_{normal}$ values of the HPR patients with the CR cases. (**D, E**) CXCL13 expression was detected by immunohistochemistry (IHC) in HPR and CR patients (**D**), and the immunoreactivity score was calculated (**E**). (**F**) Western blot analyses of lysates from the tumors and adjacent normal lung tissues harvested from CR NSCLCs. (**G**) The concentrations of CXCL13 in the blood samples from healthy donors (HDs) and HPR and CR patients were detected by ELISA. (**H, I**) *CXCL13* expression in Oncomine reports. (**H**) *CXCL13* expression was detected by microarrays in tumor samples and normal lung tissues. AD, adenocarcinoma; A+S, adenocarcinoma and squamous cell carcinoma; Ca, Canada; It, Italy; Jp, Japan; SC, squamous cell carcinoma; Tw, Taiwan, China. (**I**) The expression of *CXCL13* was detected in tumor tissues of smokers and non-smokers. (**J**) In mouse Gene Expression Omnibus (GEO) data sets, the expression of *cxcl13* in indicated mice was detected

*Figure 1 continued on next page*

*Figure 1 continued*

by microarray. (**K**) The relationship between the CXCL13 expression and the tumor stages of lung cancer patients. (**L**) Overall survival of 54 CR patients (see *Table 3* for their baseline demographic characteristics). The median follow-up was 1087 days (range, 187–1845 days).

The following source data and figure supplement are available for figure 1:

**Source data 1.** Sequences of primers for real-time PCR and ChIP, and siRNA.

**Figure supplement 1.** Kaplan–Meier estimates of survival of patients with non-small cell lung cancer (NSCLC) according to age, cancer stage, and histology.

non-smokers (27/60, 45%, p=0.04; *Table 1*), suggesting a potential association between tobacco smoke and *CXCL13* expression. The multivariate logistic analyses showed that among the 201 NSCLCs, CXCL13-high was associated with HPR (p=4.6×10$^{-6}$) and tobacco smoke (p=0.032; *Table 2*).

To investigate *CXCL13* expression in NSCLCs of other cohorts, a cancer microarray database Oncomine (*Rhodes et al., 2004*) (www.oncomine.org) was applied. We found that in several works of this database (*Okayama et al., 2012*; *Bhattacharjee et al., 2001*; *Hou et al., 2010*; *Landi et al., 2008*; *Selamat et al., 2012*; *Talbot et al., 2005*; *Su et al., 2007*; *Stearman et al., 2005*), *CXCL13* in tumor samples was elevated compared with their paired normal lung tissues or other normal controls (*Figure 1H*). *CXCL13* was also higher in smoker NSCLCs than non-smoker patients in some studies (*Okayama et al., 2012*; *Landi et al., 2008*; *Selamat et al., 2012*) (*Figure 1I*). In microarray data sets GSE6135 (*Ji et al., 2007*), GSE21581 (*Carretero et al., 2010*), and GSE54353 (*Xu et al., 2014*) deposited in the Gene Expression Omnibus (GEO; http://www.ncbi.nlm.nih.gov/geo/) from genetically engineered mouse models of lung cancer, *Cxcl13* was increased in *Kras*$^{G12D}$*Stk11*$^{-/-}$ mice

**Table 1.** Baseline demographic characteristics of the 201 patients who underwent CXCL13 analyses.

| Characteristics | Total | | | Highly polluted region (HPR) | | | Control region (CR) | | | p values (HPR vs CR) |
|---|---|---|---|---|---|---|---|---|---|---|
| | Case, n | CXCL13 high, n (%) | p values | Case, n | CXCL13 high, n (%) | p values | Case, n | CXCL13 high, n (%) | p values | |
| Total | 201 | 134 (66.7) | | 70 | 63 (90) | | 131 | 71 (54.2) | | 0.0000003 |
| G: Male | 134 | 88 (65.7) | 0.67 | 47 | 41 (87.2) | 0.27 | 87 | 47 (54) | 0.95 | 0.0001 |
| Female | 67 | 46 (68.7) | | 23 | 22 (95.7) | | 44 | 24 (54.5) | | 0.0006 |
| A: <65 y | 140 | 98 (70) | 0.21 | 56 | 51 (91.1) | 0.55 | 84 | 47 (56) | 0.7 | 0.000009 |
| ≥65 y | 56 | 34 (60.7) | | 14 | 12 (85.7) | | 42 | 22 (52.4) | | 0.03 |
| Unknown | 5 | 2 (40) | | | | | 5 | 2 (40) | | |
| S: Smoker | 107 | 75 (70.1) | 0.18 | 36 | 31 (86.1) | 0.17 | 71 | 44 (62) | 0.04 | 0.01 |
| Non-smoker | 87 | 53 (60.9) | | 27 | 26 (96.3) | | 60 | 27 (45) | | 0.000006 |
| Unknown | 7 | 6 (85.7) | | 7 | 6 (85.7) | | | | | |
| H: Adenocarcinoma (AD) | 131 | 91 (69.5) | 0.45 | 48 | 44 (91.7) | 0.37 | 83 | 47 (56.6) | 0.84 | 0.00003 |
| Squamous cell carcinoma (SCC) | 61 | 39 (63.9) | | 19 | 16 (84.2) | | 42 | 23 (54.8) | | 0.03 |
| Others | 9 | 4 (44.4) | | 3 | 3 (100) | | 6 | 1 (16.7) | | |
| Tumor node metastasis (TNM): I | 88 | 51 (58) | 0.007 | 29 | 23 (79.3) | 0.02 | 59 | 28 (47.5) | 0.05 | 0.004 |
| II | 27 | 17 (63) | | 9 | 8 (88.9) | | 18 | 9 (50) | | 0.005 |
| III | 58 | 43 (74.1) | | 17 | 17 (100) | | 41 | 26 (63.4) | | 0.004 |
| IV | 22 | 19 (86.4) | | 11 | 11 (100) | | 11 | 8 (72.7) | | 0.06 |
| Unknown | 6 | 4 (66.7) | | 4 | 4 (100) | | 2 | 0 (0) | | |

G, gender; A, age; S, smoke; H, histology.

**Table 2.** Multivariate logistic analyses of the association between CXCL13 high expression and clinical characteristics.

Highly polluted region (HPR) patients, n=70

| Variable | Odds ratio | 95.0% confidence interval | P value |
| --- | --- | --- | --- |
| Age | 1.483 | 0.177–12.452 | 0.716 |
| Gender | 4.711 | 0.488–45.51 | 0.18 |
| Smoking | 0.652 | 0.115–3.682 | 0.628 |
| Histology | 0.37 | 0.069–1.992 | 0.247 |
| TNM stage | 3.092 | 0.765–12.502 | 0.113 |

Control region (CR) patients, n=131

| Variable | Odds ratio | 95.0% confidence interval | P value |
| --- | --- | --- | --- |
| Age | 0.914 | 0.455–1.832 | 0.799 |
| Gender | 2.15 | 0.772–5.993 | 0.143 |
| Smoking | 0.513 | 0.251–1.052 | 0.06 |
| Histology | 1.148 | 0.569–2.313 | 0.7 |
| TNM stage | 2.355 | 1.157–4.793 | 0.018 |

Total (HPR and CR patients), n=201

| Variable | Odds ratio | 95.0% confidence interval | P value |
| --- | --- | --- | --- |
| Region | 7.908 | 3.272–19.114 | $4.6\times10^{-6}$ |
| Age | 0.964 | 0.5–1.861 | 0.914 |
| Gender | 2.254 | 0.897–5.662 | 0.084 |
| Smoking | 0.394 | 0.168–0.925 | 0.032 |
| Histology | 0.996 | 0.52–1.907 | 0.99 |
| TNM stage | 2.707 | 1.401–5.232 | 0.003 |

(especially in metastatic tumors), as compared with $Kras^{G12D}Stk11^{wt}$ mice (*Figure 1J*). These results suggest that *CXCL13* overexpression was not specific to the Chinese cohorts, and *Cxcl13* may play a role in *Stk11*-related lung tumorigenesis.

We showed patients of both HPR and CR regions with relatively earlier disease (stages I and II) had lower *CXCL13*, while those with advanced disease (stages III and IV) had higher *CXCL13* (*Figure 1K*), and multivariate logistic analyses showed that CXCL13-high was associated with TNM stage (*Table 2*, p=0.003). In 54 CR patients whose survival information was available (*Table 3*), the median survival time of CXCL13-high patients (965 days) was much shorter than the CXCL13-low cases (1193 days, p=0.03; *Figure 1L*). Kaplan-Meier estimates of survival of patients with NSCLC according to age (*Figure 1—figure supplement 1A*), cancer stage (*Figure 1—figure supplement 1B*), and histology (*Figure 1—figure supplement 1C*) confirmed that patients with stages III–IV lung cancer had shorter survival time than those with earlier stages of NSCLCs.

## BaP induces CXCL13 production in vitro and in vivo

PAHs were reported to be the major carcinogens in the $PM_{2.5}/PM_{10}$ in HPR, as well as in the PM at urban locations in Beijing, Shanghai, Guangzhou and Xi'an in China during January 2013 (*Huang et al., 2014*). Clinically, a long latency is required for individuals to develop lung cancer since they were first exposed to smoking or air pollution. To test the effects of PAHs on cytokine/chemokine production, the normal human lung epithelial 16HBE cells (*Cozens et al., 1994*) were exposed to a representative PAH compound benzo(a)pyrene (BaP) at 1 μM for a long period of time (30 days). We found that *CXCL13* was the most significantly up-regulated gene among the 84 cytokines/chemokines (*Figure 2A*). We confirmed that BaP up-regulated CXCL13 at both the mRNA and protein levels in a dose- and time-dependent manner in 16HBE and A549 lung cancer cells (*Figure 2B, C*).

**Table 3.** Baseline demographic characteristics of 54 control region (CR) lung cancer patients whose survival information was available.

| Characteristics | Case, n | CXCL13-high, n (%) | P* Value |
|---|---|---|---|
| Total | 54 | 25 (46.3) | |
| Gender | | | |
| Male | 38 | 16 (42.1) | 0.34 |
| Female | 16 | 9 (56.3) | |
| Smoking | | | |
| Smoker | 33 | 16 (48.5) | 0.69 |
| Non-smoker | 21 | 9 (42.9) | |
| Age, years | | | 0.72 |
| <65 | 38 | 17 (44.7) | |
| ≥65 | 16 | 8 (50) | |
| Histology | | | |
| Adenocarcinoma | 32 | 15 (46.9) | 0.83 |
| squamous cell carcinoma | 18 | 9 (50) | |
| small cell lung cancer | 1 | 0 (0) | |
| Others | 3 | 1 (33.3) | |
| TNM stage | | | 0.65 |
| I | 26 | 12 (46.2) | |
| II | 6 | 2 (33.3) | |
| III | 18 | 9 (50) | |
| IV | 4 | 2 (50) | |

To test the in vitro observations in vivo, A/J mice were treated with vehicle control (corn oil) or 50–100 mg/kg BaP (*Figure 2—figure supplement 1A*, modified from previous studies (*Wattenberg and Estensen, 1996*). All BaP-treated animals developed lung cancer within 6 months, as detected by a microCT scan (*Figure 2—figure supplement 1B,C*) and histopathology (*Figure 2D*). We found that Cxcl13 was significantly up-regulated in the tumor samples compared with their normal counterparts at both the mRNA (*Figure 2E*) and protein (*Figure 2D*) levels, which paralleled the increase in Cxcl13 serum concentrations (*Figure 2F*). Notably, BaP increased Cxcl13 concentrations in peripheral blood of the mice at 1 month after the beginning of the treatment (*Figure 2F*), at which point no tumors were observed (*Figure 2—figure supplement 1D*).

To determine the source of Cxcl13, we performed IHC and immunofluorescence assays in lung cancer tissues of the mice. To do this, thyroid transcription factor-1 (Ttf-1) which is expressed in the lung epithelial cells (*Lazzaro et al., 1991*), was used to mark lung cancer cells, and Cd68 stain was employed to mark macrophages which can produce Cxcl13 in inflammatory lesions with lymphoid neogenesis (*Carlsen et al., 2004*). By IHC assay, we found that both the Ttf1-positive lung cancer cells and Cd68-positive macrophages were stained positive for Cxcl13, but Ttf1-positive cells constituted more than 95% of the cellular component of the lung tumor tissues of the mice (*Figure 2G*). This observation was confirmed by immunofluorescence assay (*Figure 2H*) using antibodies against Cxcl13 (green), Cd68 (red), and Ttf1 (white). These results indicate that Ttf1 positive lung cancer cells were the main source of Cxcl13 in mice exposed to BaP.

We further showed that the anti-inflammatory drug dexamethasone (DEX) (*Wattenberg and Estensen, 1996*) down-regulated Cxcl13 mRNA and protein, and reduced the Cxcl13 serum concentration (*Figure 2D–F*). DEX reduced the tumor burden (*Figure 2—figure supplement 1B,C*, *Figure 2D*) and prolonged the survival (*Figure 2I*) of the mice. Cxcl12, which was shown to be associated with lung cancer (*Teicher and Fricker, 2010*), was not significantly changed in the BaP-treated mice (*Figure 2—figure supplement 1E*).

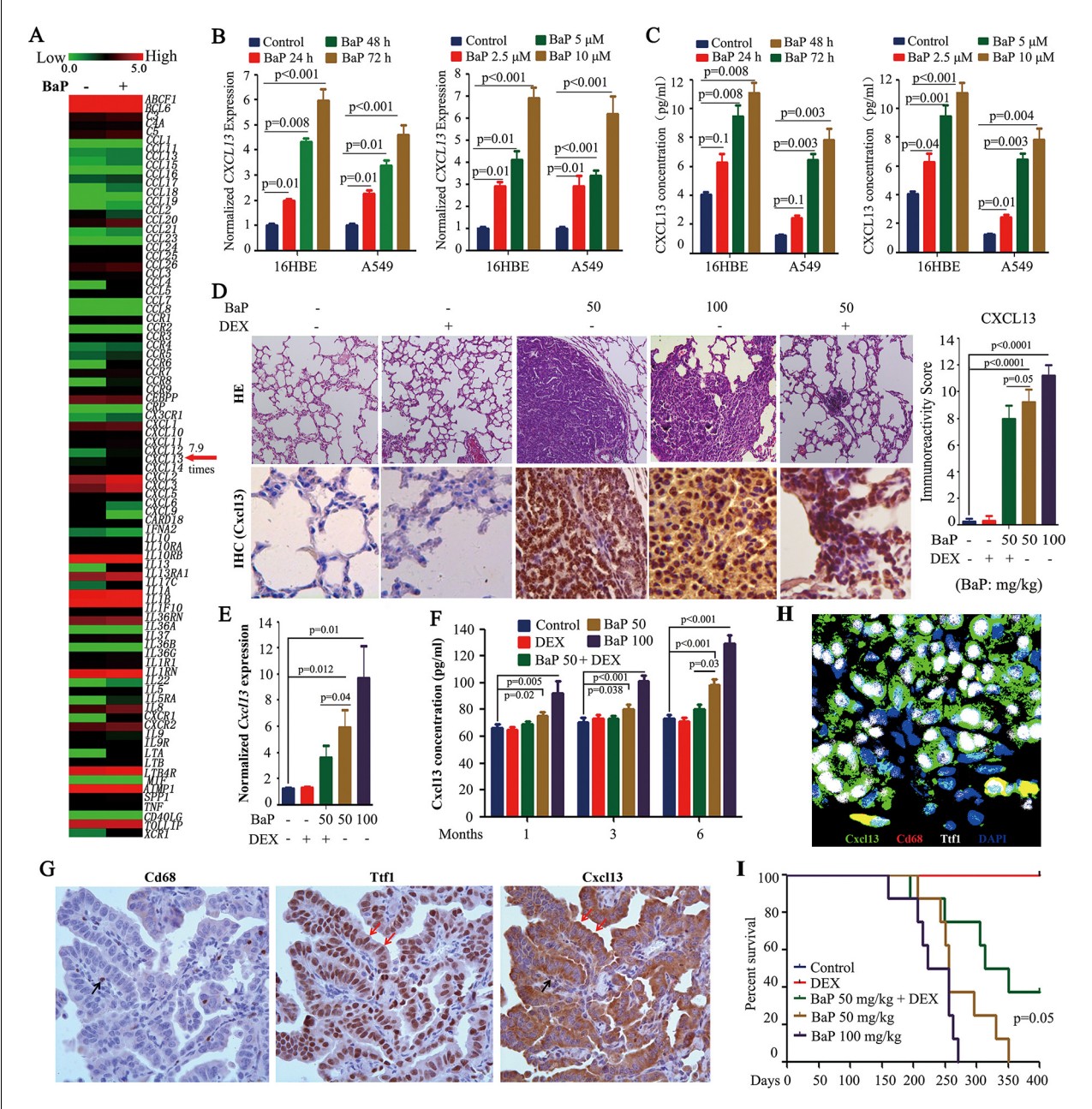

**Figure 2.** Benzo(a)pyrene (BaP) induces CXCL13 in vitro and in vivo. (**A**) A PCR array analysis of the expression of 84 cytokines/chemokines in 16HBE normal lung epithelial cells treated with 1 μM BaP for 30 days. (**B**) The cells were treated with BaP at 10 μM for indicated time points or with the indicated concentrations for 72 hr, and *CXCL13* expression was assessed by real-time RT-PCR. The experiments were conducted in triplicate and repeated three times. The error bars represent the SD. (**C**) The cells were treated with BaP as described in (**B**), and the concentration of CXCL13 in the supernatants was evaluated by ELISA. (**D**) The A/J mice were treated with BaP and/or dexamethasone (DEX) for 5 weeks (see also *Figure 2—figure supplement 1A*) and sacrificed 6 months later. The lung tissues were isolated and analyzed by Hematoxylin and eosin (HE) staining or immunohistochemistry (IHC) using an anti-Cxcl13 antibody (left panel). The immunoreactivity score was calculated (right panel). (**E**) *Cxcl13* expression was detected in the lung tissues by real-time PCR. (**F**) The concentration of Cxcl13 in mouse serum was assayed by ELISA. (**G**) IHC assays of mice' lung tumor tissues using anti-Cd68, anti-Ttf1, and anti-Cxcl13 antibodies. (**H**) Immunofluorescence assay of mice' lung tumor tissues using antibodies against Cxcl13 (green), Cd68 (red), and Ttf1 (white). 4',6-diamidino-2-phenylindole (DAPI) was used to stain the nucleus (blue). (**I**) The survival curves of the mice treated with BaP and/or DEX (n=8 for each group).

The following figure supplement is available for figure 2:

**Figure supplement 1.** Benzo(a)pyrene (BaP) induces lung cancer in A/J mice.

## BaP induces CXCL13 production by facilitating AhR translocation to the nucleus

We screened for transcription factors that could regulate *CXCL13*, and found potential aryl hydrocarbon receptor (AhR) (*Figure 3A*) and V-Rel Avian Reticuloendotheliosis Viral Oncogene Homolog

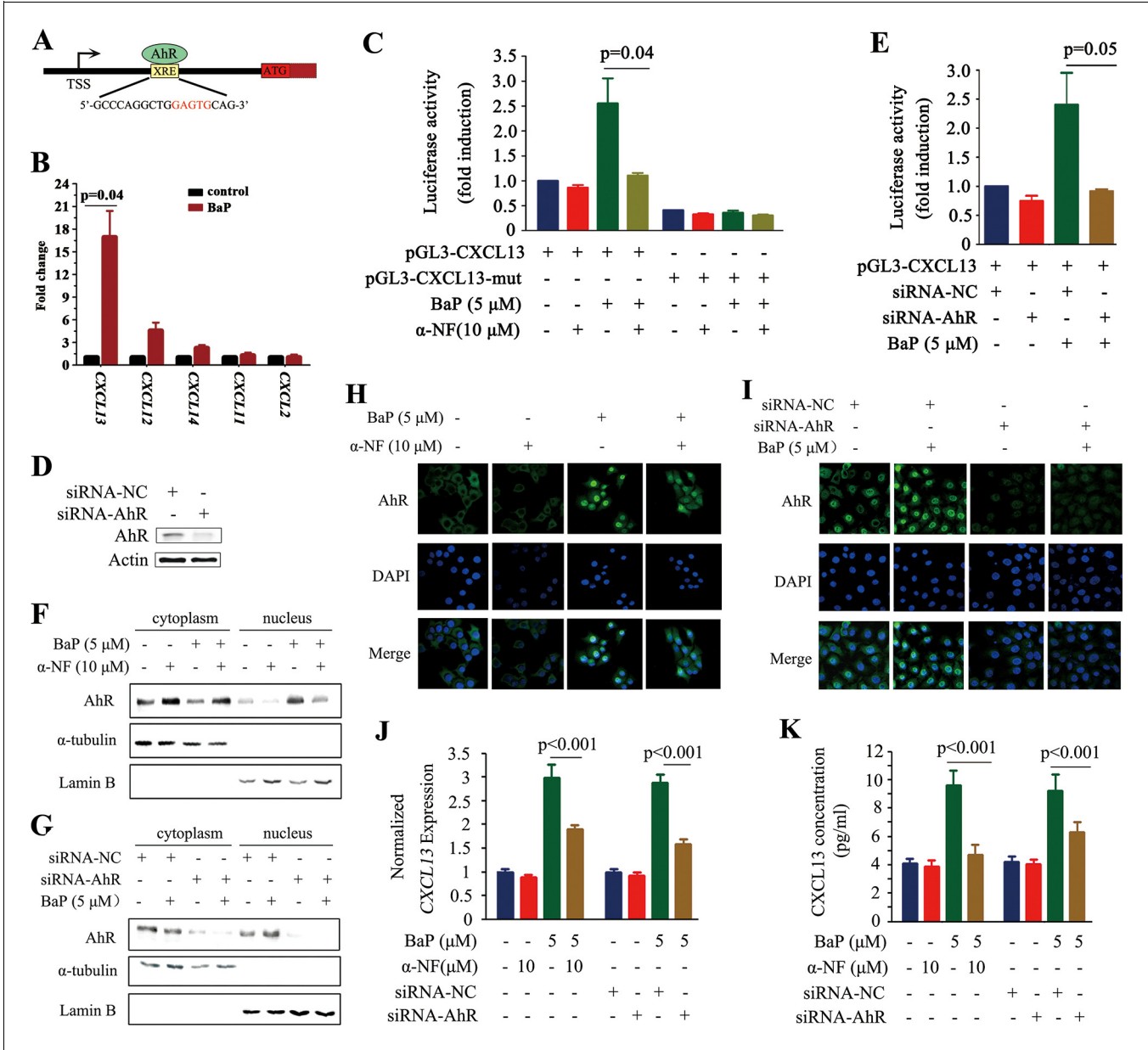

**Figure 3.** *CXCL13* is a target gene of aryl hydrocarbon receptor (AhR). (A) The AhR binding site is located at 1.7 kb downstream of the *CXCL13* transcription start site (TSS). (B) A chromatin immunoprecipitation (ChIP) assay was performed in BaP-treated or untreated 16HBE cells. The enriched *CXCL13* was detected by qPCR. (C) The A549 cells were transfected with the wild-type (WT) or mutant (deletion mutation (mut) in the XRE-like sequence) *CXCL13* promoter-luciferase reporter construct, treated with BaP and/or α-NF for 48 hr, and assessed by the luciferase assays. (D, E) A549 cells were transfected with AhR-specific siRNAs, and western blot was performed to detect the expression of AhR. Three siRNAs were used, and the result of one was shown (D). Luciferase assays were performed in A549 cells transfected with the WT *CXCL13* promoter-luciferase reporter construct and siRNAs in the absence or presence of BaP (E). (F, G) Western blot analyses of AhR in the cytoplasm and nucleus of 16HBE cells co-incubated with BaP, with or without α-NF treatment (F) or siRNA transfections (G). (H, I) Immunofluorescence assays of AhR expression in 16HBE cells co-incubated with BaP, with or without α-NF treatment (H) or siRNA transfections (I). (J, K) CXCL13 mRNA (detected by qPCR; J) and protein (in supernatants of the cells detected by ELISA; K) levels in the AhR-silenced 16HBE cells treated with BaP and/or α-NF.

A (RelA) (data not shown) binding sites in its promoter. AhR is a ligand-activated transcription factor that binds the xenobiotic-responsive element (XRE) or aryl hydrocarbon response element (AHRE) to regulate genes in response to the planar aromatic hydrocarbon 2,3,7,8-tetrachlorodibenzo-p-dioxin (TCDD) (*Fujisawa-Sehara et al., 1987*; *Lo and Matthews, 2012*; *Shimizu et al., 2000*). The potential XRE-like sequence, 5'-GCCCAGGCTGGAGTGCAG-3', is located at 1.7 kb downstream from the transcription start site (TSS; *Figure 3A*). The potential interaction between AhR and *CXCL13* was analyzed by the chromatin immunoprecipitation (ChIP) assay, and the results showed that AhR could not bind *CXCL13* in A549 cells without BaP treatment. Interestingly, AhR-*CXCL13* interaction was detected and *CXCL13* was significantly up-regulated in the presence of BaP, revealed by quantitative PCR (qPCR) (*Figure 3B*). However, significant interaction between AhR and *CXCL12, CXCL14, CXCL11*, or *CXCL2* was not detected (*Figure 3B*). BaP significantly induced luciferase activity driven by the *CXCL13* promoter containing XRE-like sequence; deletion of the XRE-like element, co-incubation with the AhR antagonist α-NF (*Wilhelmsson et al., 1994*) or transfection of an AhR-specific siRNA significantly reduced luciferase activity (*Figure 3C–E*).

TCDD can activate AhR by inducing its translocation from the cytoplasm to nucleus (*Pollenz et al., 1994*). We reported that α-NF and siAhR drastically inhibited BaP caused translocation of AhR to nucleus in 16HBE cells (*Figure 3F–I*). These results might explain why α-NF and siAhR significantly decreased *CXCL13* expression (*Figure 3J*) and concentration in the supernatant (*Figure 3K*) in BaP-treated 16HBE cells.

## Knockdown of Cxcl13 attenuates BaP-induced lung cancer

To uncover the role of CXCL13 in BaP-induced lung cancer, *Cxcl13* knockout mice (*Cyster et al., 2000*) were treated with 100 mg/kg BaP twice a week for 8 weeks (*Figure 4—figure supplement 1A,B*), and the tumor burden of the mice was evaluated. First, tumor nodules in histologic sections of mice upon BaP treatment were analyzed as described (*Tan et al., 2013*), and the results showed that at treatment time points of 120 days, 180 days, and 240 days, $Cxcl13^{-/-}$ mice had much fewer lesions than $Cxcl13^{+/-}$ and $Cxcl13^{+/+}$ mice (*Figure 4A*). Then, microCT was used to detect visible tumors in mice 240 days after BaP treatment. We found that $Cxcl13^{-/-}$ mice harbored much fewer lung tumors than $Cxcl13^{+/-}$ and $Cxcl13^{+/+}$ mice (*Figure 4B*). Consistent with this observation, the tumor volume of $Cxcl13^{-/-}$ mice was significantly smaller than $Cxcl13^{+/-}$ and $Cxcl13^{+/+}$ mice (*Figure 4C*). In $Cxcl13^{+/+}$ and $Cxcl13^{-/+}$ mice, the Cxcl13 serum concentrations were elevated 3 months after BaP treatment, at which time point no tumors were detected (*Figure 4D*). Moreover, the life span of BaP-treated $Cxcl13^{-/-}$ mice was prolonged compared with the $Cxcl13^{+/+}$ and $Cxcl13^{-/+}$ mice (*Figure 4E*). These results indicate that CXCL13 is required for BaP-induced lung cancer.

## CXCL13-CXCR5 signaling is critical for BaP-induced lung cancer

CXCL13 primarily functions by binding to the G protein coupled receptor CXCR5, and CXCL13 is currently the only known ligand of CXCR5 (*Förster et al., 1996*; *Lazennec and Richmond, 2010*). We tested the expression of CXCR5 in 24 NSCLCs, and found that it was expressed by the tumor samples but was not significantly higher than in paired normal lung tissues (p=0.089; *Figure 4F*). In A/J mice treated with BaP, *Cxcr5* was slightly up-regulated (p=0.07; *Figure 4G*). To further show the role CXCR5 plays in lung carcinogenesis, $Cxcr5^{-/-}$ mice (*Förster et al., 1996*) were treated with BaP at 100 mg/kg BaP twice a week for 8 weeks (*Figure 4—figure supplement 1B,C*). We showed that BaP treatment induced lung cancer in $Cxcr5^{+/+}$ mice (*Figure 4H,I*). However, BaP-treated $Cxcr5^{+/-}$ mice developed fewer tumors, and BaP-treated $Cxcr5^{-/-}$ mice had much ewer lung tumors (*Figure 4H,I*). Consistently, the tumor volume of $Cxcr5^{-/-}$ mice was significantly lower than $Cxcr5^{+/-}$ and $Cxcr5^{+/+}$ mice (*Figure 4J*). Furthermore, $Cxcr5^{-/-}$ mice had a prolonged life span compared with the wild-type and $Cxcr5^{+/-}$ mice (*Figure 4K*). These results demonstrate that CXCR5 is also critical for BaP-induced lung cancer.

## CXCR5-expressing macrophages promote lung cancer progression

CXCR5 is expressed by B cells, T cells and macrophages (*Förster et al., 1996*; *Nigrovic and Lee, 2005*). We investigated which cells were involved in BaP-induced lung cancer by analyzing cell surface markers in the tumor microenvironment using antibodies against CD45, CD19, CD4, and CD68.

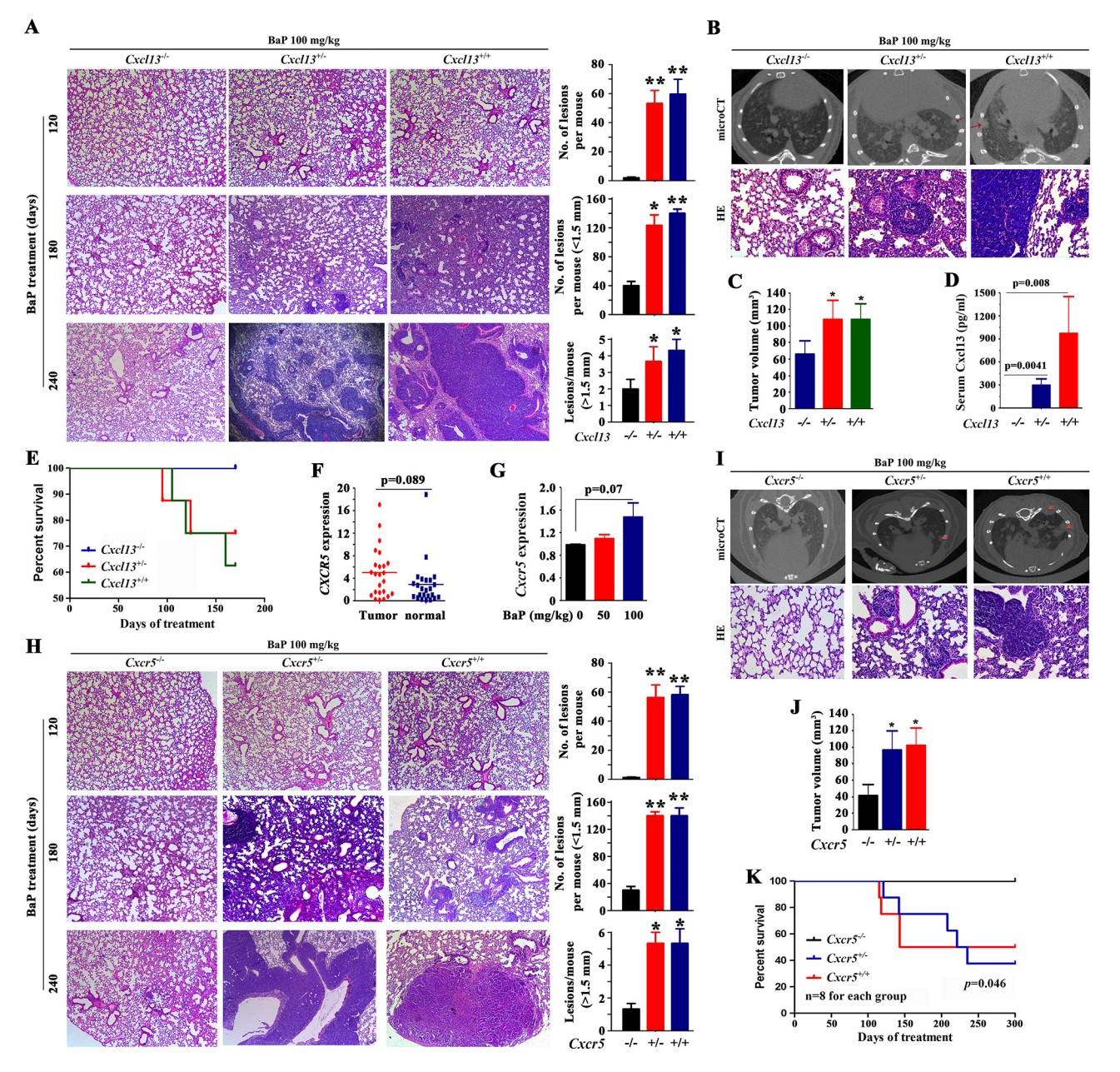

**Figure 4.** Cxcl13 and Cxcr5 are critical to benzo(a)pyrene (BaP)-induced lung cancer. (A) *Cxcl13* deficiency mice were treated with BaP, sacrificed 120 days, 180 days or 240 days later, and the tumor nodules in histologic sections were analyzed. See also (*Figure 4—figure supplement 1*). (B) MicroCT scanning images and HE staining of lung sections from the BaP-treated *Cxcl13* wild-type (WT) or knockout mice. (C) Tumor volume of the microCT scanning of the mice. (D) Serum concentrations of Cxcl13 in the BaP-treated *Cxcl13* WT or knockout mice. (E) Life span of the BaP-treated *Cxcl13⁺/⁺*, *Cxcl13⁺/⁻*, and *Cxcl13⁻/⁻* mice. (F, G) *Cxcr5* expression in non-small cell lung cancers (NSCLCs, n=24; F) and in A/J mice treated with BaP (n=6 for each group; G). (H) *Cxcr5* deficiency mice were treated with BaP, sacrificed 120 days, 180 days or 240 days later, and the tumor nodules in histologic sections were analyzed. See also *Figure 4—figure supplement 1*. (I) MicroCT scanning images and HE staining of lung sections from BaP-treated *Cxcr5* WT or knockout mice. (J) Tumor volume of the microCT scanning of the mice. (K) Life span of the BaP-treated Cxcr5 WT or knockout mice. *p<0.05; **p<0.01.

The following figure supplement is available for figure 4:

**Figure supplement 1.** Treatment of *Cxcl13⁻/⁻* and *Cxcr5⁻/⁻* mice with benzo(a)pyrene (BaP).

Flow cytometry analysis showed a significant increase in Cd68[+] macrophages in the lung tumor samples from BaP-treated A/J mice (*Figure 5A,B*), which was confirmed by IHC (*Figure 5C*). We found that 94.7% of these macrophages strongly expressed Cxcr5 (*Figure 5D*). However, B cells and T cells were not enriched in the tumor microenvironment (*Figure 5—figure supplement 1*). Moreover, CD68[+] macrophages were enriched in human lung tumor tissues revealed by IHC (*Figure 5C*) and immunofluorescence assays (*Figure 5E*). The Tamm–Horsfall protein (THP-1) macrophage cell line also expressed high levels of CXCR5 and CD68 (*Figure 5E*).

We investigated the potential role for these macrophages in promoting lung cancer. In a transwell migration assay, THP-1 cells induced the migration of A549-Luciferase (Luc) cells and H1975 cells from the upper chamber to the lower chamber (*Figure 5F*). Stable transfection of *CXCL13* into A549-Luc cells (designated A549-Luc-CXCL13 cells) significantly enhanced the migration activity, which was antagonized by an anti-CXCR5 antibody (*Figure 5F*). Addition of CXCL13 to THP-1 cells enhanced the migration of H1975 cells, while an anti-CXCR5 antibody inhibited this effect (*Figure 5F*).

A549-Luc-CXCL13 cells were injected into the right lung of the non-obese diabetic/severe combined immunodeficiency (NOD/SCID) mice (whose macrophages were functionally immature), and bioluminescence was recorded 30 days later. We showed that compared with the A549-Luc cells, the A549-Luc-CXCL13 cells significantly increased the tumor burden in the recipient mice (*Figure 5G,H*). Furthermore, tail vein-injection of THP-1 cells increased the volume of tumors formed by the A549-Luc-CXCL13 cells (*Figure 5G,H*) and caused obvious metastases to the left lung of the mice (*Figure 5I*).

## CXCL13 induces SPP1 production by macrophages

The above results suggested that CXCL13 may induce macrophages to produce critical factors that promote lung cancer progression/metastasis. To identify these factors, a whole genome gene expression assay was performed in the eight patients (*Figure 1A*), and 436 genes were found to be associated with *CXCL13* (a coefficient >0.5 or <−0.5; *Figure 6—source data 1*). These genes were enriched in cytokine-cytokine receptor interactions, the Wnt signaling pathway, the calcium signaling pathway, and others (*Figure 6A*). Among the secretion-related genes, *MMP12, secreted phosphoprotein 1 (SPP1)*, which encodes a macrophage-secreted cytokine SPP1 or osteopontin (*Kohri et al., 1992*), and *MMP7* had the highest coefficient values for interacting with *CXCL13* (*Figure 6B*). We tested 11 secretion-related genes in THP-1 cells, and found that the expression of *SPP1* was the highest (*Figure 6C*). Therefore, it was chosen for further investigation. In THP-1 cells, CXCL13 treatment significantly increased SPP1 concentrations in the supernatant (*Figure 6D*). In trans-well assays, the addition of CXCL13 to THP-1 cells in the lower chamber enhanced the migration of H1975 cells, while SPP1 silencing by siSPP1 in THP-1 cells attenuated this effect (*Figure 6E*, left). Transfection of CXCL13 into A549 cells increased their migration activity, while SPP1 silencing in THP-1 cells inhibited A549-CXCL13 cell migration (*Figure 6E*, right panel). In BaP-treated A/J mice, the Spp1 serum concentration was significantly increased (*Figure 6F*), and the tumor samples showed significantly increased Spp1 IHC staining compared with the control mice (*Figure 6G*). By IHC assay, we found that Cd68 positive macrophages were also strongly stained by Spp1, whereas Ttf1-positive lung cancer cells were only weakly stained positive for Spp1 (*Figure 6H*). This observation was confirmed by immunofluorescence assay (*Figure 6I*) using antibodies against Spp1 (green), Cd68 (red), and Ttf (white). These results indicate that Cd68-positive macrophages were the main source of Spp1 in mice exposed to BaP.

## CXCL13-CXCR5-SPP1 signaling induces an EMT phenotype

At least two tumors were found in left and/or right lungs of each BaP-treated mouse (*Figure 2—figure supplement 1B*), and CXCL13 induced cancer cell migration and metastasis (*Figure 5F–I*, *Figure 6E*). We tested the epithelial mesenchymal transition (EMT) phenotypes that are associated with cancer progression and metastasis (*Kalluri and Weinberg, 2009*) in tumor samples from the BaP-treated A/J mice, and found that the expression of *E-Cadherin* was down-regulated, while *N-Cadherin, Vimentin, Slug* and *Snail* were up-regulated (*Figure 6—figure supplement 1A*). At protein level, E-Cadherin was down-regulated, while N-Cadherin and Vimentin were up-regulated in the tumors (*Figure 6—figure supplement 1B*). β-catenin transport to the nucleus is critical for cells to

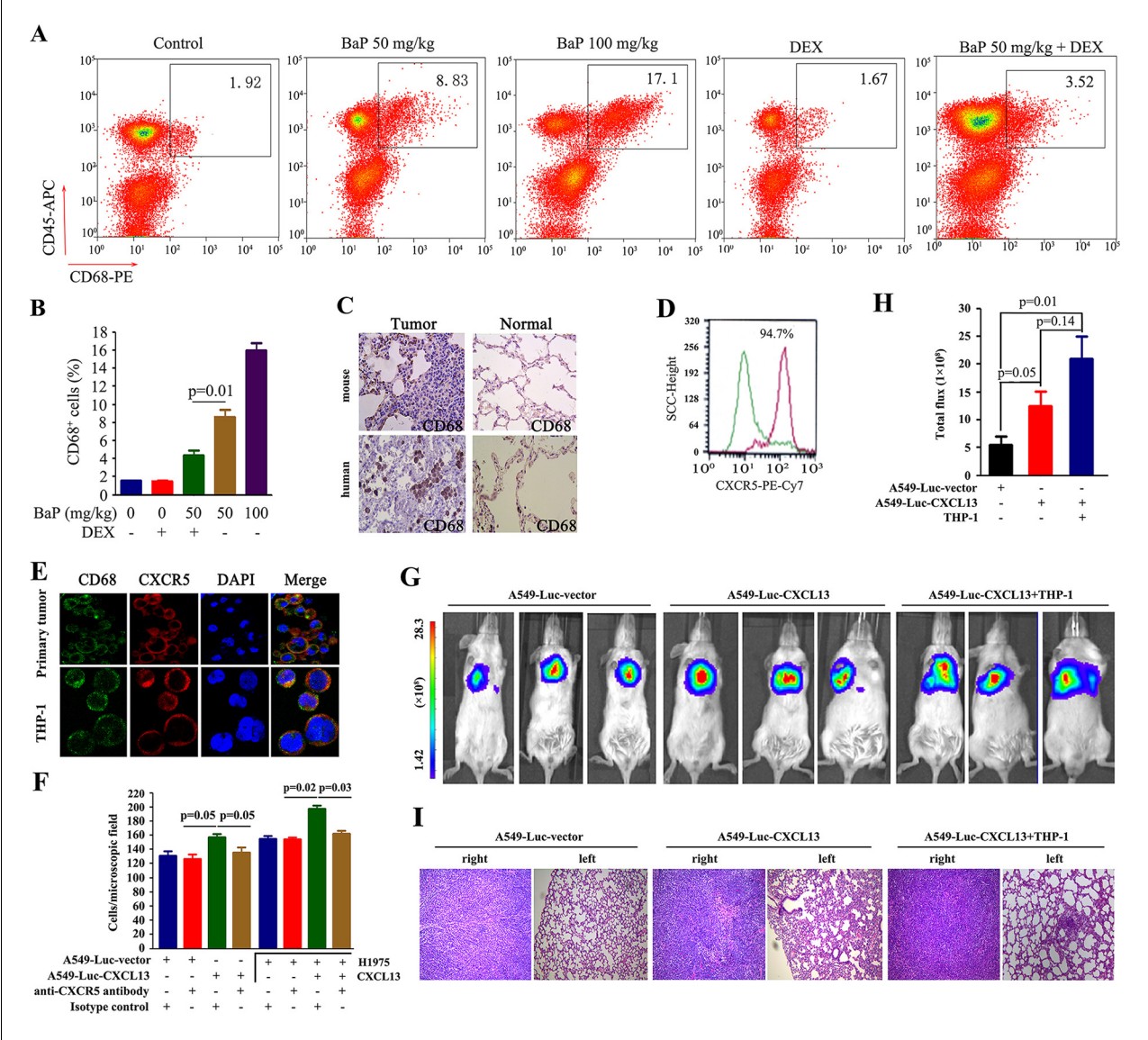

**Figure 5.** Tumor-associated macrophages in benzo(a)pyrene (BaP)-induced lung cancer. (**A, B**) Flow cytometry analysis of Cd68[+] macrophages in BaP-induced tumors. A representative gating is shown. The numbers indicate the Cd68[+] cells in the quadrant expressed as the percentage of the total Cd45 + leukocytes from the same tumor (**A**). The means+SD of the Cd68[+] cells from the mice (n=10 for each group) are shown (**B**). See also *Figure 5—figure supplement 1*. (**C**) IHC analysis of CD68[+] macrophages in tumor samples from BaP-treated mice and highly polluted region (HPR) patients. (**D**) Flow cytometry analysis of Cd68[+] macrophages isolated from tumor samples of mice treated with 50 mg/kg BaP using an anti-Cxcr5 antibody. (**E**) Immunofluorescence analysis of tumor-associated macrophages in tumor samples from HPR patients and THP-1 cells using anti-CD68 and anti-CXCR5 antibodies; DAPI was used to counterstain the nucleus. (**F**) A trans-well migration assay was performed by plating THP-1 cells in the lower chambers, and the indicated cells in the upper chambers, with or without anti-CXCR5 antibody. (**G, H**) Bioluminescent assays of mice that were inoculated with A549- Luciferase (Luc) or A549-Luc-CXCL13 cells ($8\times10^5$) in the right lung. THP-1 cells ($8\times10^5$) were injected via the tail vein. Representative images (**G**) and total luminous flux (**H**) were shown. (**I**) Lung sections of the mice were stained with HE.

The following figure supplement is available for figure 5:

**Figure supplement 1.** Analysis of B cells and T cells in benzo(a)pyrene (BaP)-induced lung cancer.

enter into an EMT and acquire an invasive phenotype (*Kim, 2002*). We showed that the nuclear β-catenin levels were increased in the tumor samples from A/J mice (*Figure 6—figure supplement 1C*). In the tumor samples from NOD/SCID mice injected with A549-Luc-CXCL13 cells (*Figure 6—*

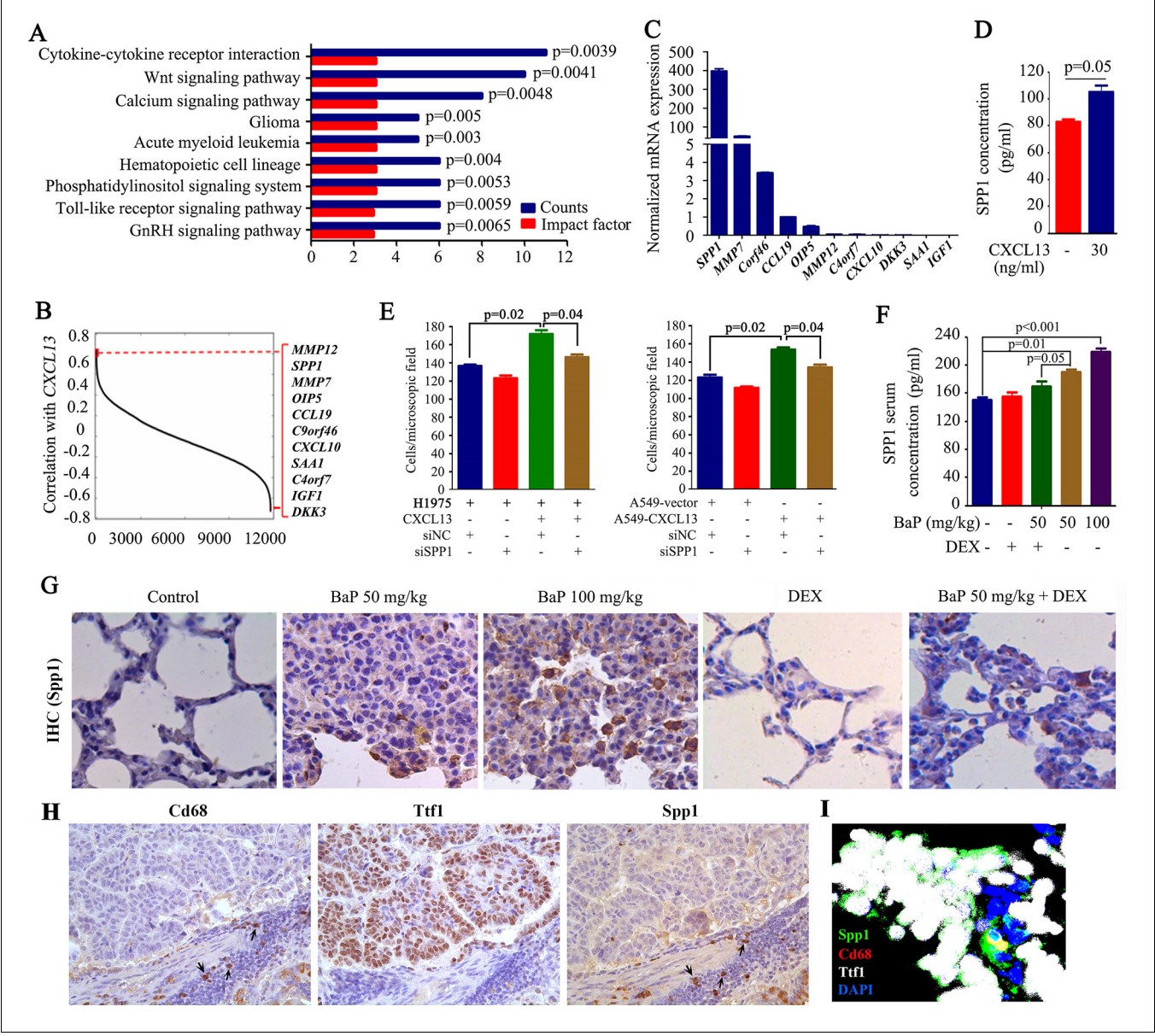

**Figure 6.** Identification of macrophage-secreted SPP1 as a downstream effector of CXCL13. (**A**) The pathway analysis of *CXCL13* associated genes. The data from the microarray data sets of the eight highly polluted region (HPR) lung cancers are shown. See also *Figure 6—source data 1*. (**B**) Gene ranking according to the correlation with *CXCL13* expression. The genes were filtered based on extracellular localization to identify paracrine mediators. The list on the right shows the genes that correlate most significantly with *CXCL13*. (**C**) The mRNA expression of the candidate gene was detected in THP-1 cells by qPCR. (**D**) The concentration of SPP1 in supernatants of THP-1 cells in the absence or presence of CXCL13. (**E**) Trans-well migration assays were performed by plating THP-1 cells (transfected with siSPP1 or siNC) in the lower chambers, and the indicated cells (CXCL13-treated or untreated) in the upper chambers. (**F**) Spp1 serum concentrations of mice treated with benzo(a)pyrene (BaP) and/or dexamethasone (DEX) were detected by ELISA. (**G**) Spp1 expression in lung section of mice treated with BaP and/or DEX was determined by IHC. (**H**) IHC assays using antibodies against Cd68, Ttf1, and Spp1. (**I**) Immunofluorescence assay using antibodies against Spp1 (green), Cd68 (red), and Ttf1 (white). DAPI was used to stain the nucleus (blue).

The following source data and figure supplement are available for figure 6:

**Source data 1.** *CXCL13*-associated genes in lung cancer.

**Figure supplement 1.** Activation of *β*-catenin and epithelial mesenchymal transition (EMT) in benzo(a)pyrene (BaP)-induced lung cancer.

*figure supplement 1D,E*) and BaP-treated Cxcr5$^{+/+}$ mice (*Figure 6—figure supplement 1F,G*), the expression of E-Cadherin was low, while the N-Cadherin and nuclear β-catenin were high.

SPP1 was shown to activate β-catenin (*Robertson and Chellaiah, 2010*). To study the role of SPP1 in promoting the progression of lung cancer, A549 cells were transfected with pcDNA3.1-flag-SPP1. We found that SPP1 overexpression decreased the cytoplasmic expression but increased the nuclear expression of β-catenin (*Figure 6—figure supplement 1H,I*). SPP1 overexpression in cancer cells led to the down-regulation of E-Cadherin and up-regulation of N-Cadherin and Slug at both the mRNA and protein levels (*Figure 6—figure supplement 1J,K*), and up-regulation of *Vimentin* and *Snail* at the mRNA level (*Figure 6—figure supplement 1J*). These findings were also observed in A549 cells supplemented with supernatants from THP-1 cells that were co-incubated with CXCL13 (*Figure 6—figure supplement 1H–K*). These results suggest that SPP1 may transactivate β-catenin to induce an EMT in BaP-treated mice.

## Discussion

In this study, we took the advantage of the epidemiology profile of Xuanwei lung cancer to systematically explore the abnormalities in inflammatory factors that are critical to air pollution-induced lung cancer, and reported that *CXCL13* was up-regulated in 63/70 (90%) HPR NSCLCs (*Table 1*). BaP induced lung epithelial cells to secret CXCL13 (*Figure 2*), which recruited tumor-associated macrophages (TAMs) and induced SPP1 production (*Figure 5*, *6*). SPP1 activated β-catenin by facilitating its translocation to the nucleus and promoted an EMT, leading to the progression of lung cancer (*Figure 6—figure supplement 1*). CXCL13-CXCR5 signaling was required for PAH-induced lung cancer, because knockdown of Cxcl13 or Cxcr5 attenuated BaP-induced lung cancer in mice (*Figure 4*). These results suggest that the ubiquitous carcinogen BaP may induce tumor-promoting inflammation to facilitate the invasive growth and metastasis of lung cancer, shedding new insights into the carcinogenic mechanisms of air pollution.

CXCL13 may be associated with lung cancer risk in smokers (*Shiels et al., 2013*). We showed that in HPR, smoker and non-smoker NSCLCs had approximately equal expression levels of CXCL13, suggesting a carcinogenic effect of air pollution. In CR, however, smoker patients had higher CXCL13 expression than non-smoker cases (*Figure 1* and *Table 1*). Moreover, high CXCL13 expression was associated with advanced stage cancer and poor prognosis (*Figure 2*). These results indicate that CXCL13 is a common and critical factor in air pollution-induced and tobacco smoke-induced lung carcinogenesis. Furthermore, the serum concentration of CXCL13 was elevated in mice before the emergence of detectable lung tumor by microCT (*Figure 2* and *Figure 2—figure supplement 1*), suggesting that this chemokine may represent a biomarker for early diagnosis, which warrants further investigation.

PAHs are key carcinogens in air pollution and tobacco smoke (*Huang et al., 2014*; *Hecht, 2012*), and cause mutations in *TP53* and *K-RAS*, resulting in uncontrolled cell growth (*Hecht, 2012*). PAHs also perturb the immune system and induce production of the inflammatory factors to facilitate cancer progression in cellular and animal models (*Zaccaria and McClure, 2013*; *N'Diaye et al., 2006*; *Umannová et al., 2011*; *Chen et al., 2012*; *Dreij et al., 2010*). By treating 16HBE cells with a relatively low concentration (1 µM) of BaP for a long time course (30 days) to mimic BaP-induced lung cancer in humans, we showed that *CXCL13* was the most significantly up-regulated gene among the 84 cytokines/chemokines (*Figure 2*). Furthermore, CXCL13 was increased in 134/201 (66.7%) NSCLCs (*Table 1*), suggesting the clinical relevance of CXCL13 in BaP-induced lung cancer. AhR is critical for the carcinogenic action of BaP (*Shimizu et al., 2000*). We found that CXCL13 was a direct target of AhR (*Figure 3*), and deficiency in CXCL13 or its receptor, CXCR5, abrogated BaP-induced lung cancer (*Figure 4*), while an anti-CXCR5 antibody significantly inhibited lung cancer cell migration (*Figure 5*). These results further demonstrate CXCL13's critical role in environmental pollutant-induced lung cancer, but its role in pre-malignancy and cell transformation remains to be addressed.

Cross-talk between cancer cells and cells of the neoplastic stroma is involved in the acquired ability for invasive growth and metastasis (*Hanahan and Weinberg, 2011*). Using a whole -genome gene expression array (*Figure 6*), we discovered that TAMs secreted SPP1, which activates β–catenin and is regulated by this transcription factor (*Robertson and Chellaiah, 2010*; *El-Tanani et al., 2001*). SPP1 induced the nuclear localization and activation of β–catenin in epithelial and cancer cells, resulting in an EMT phenotype (*Figure 6—figure supplement 1H–K*). At this stage, the

detailed mechanisms of CXCL13-CXCR5-induced SPP1 production were unclear. However, our results suggested that SPP1-β–catenin could form a positive feedback loop to promote EMT and lung cancer progression. CXCL13 was shown to be able to induce an EMT in breast cancer via RANKL and Src activation (*Biswas et al., 2014*). We found that BaP or CXCL13 treatment or SPP1 overexpression in lung epithelial or cancer cells did not up-regulate RANKL or Src (data not shown), but did activate β–catenin (*Figure 6—figure supplement 1*), suggesting that CXCL13 may induce an EMT via different mechanisms in different settings.

Chemokines can modulate tumor cell proliferation, survival, angiogenesis, senescence and metastasis (*Wang et al., 2015*). For example, CCL20 is induced by another important carcinogen in tobacco smoke, nicotine-derived nitrosaminoketone, and is overexpressed in smoker lung cancers and inversely associated with prognosis. CCL20 promotes lung cancer cell proliferation and migration (*Wang et al., 2015*). CXCL12 and CCR7 regulate the metastasis of breast cancers and NSCLCs (*Phillips et al., 2003*; *Müller et al., 2001*; *Takanami, 2003*), while CXCL1/2 enhance cell survival and promote chemoresistance (*Acharyya et al., 2012*). CXCL13 is overexpressed in breast cancer (*Panse et al., 2008*) and mediates prostate cancer cell proliferation (*El-Haibi et al., 2011*). Given their critical roles in cancer, pharmaceutical development pipelines are filled with new chemokine-targeting drugs to treat malignancies (*Klarenbeek et al., 2012*). DEX is a synthetic glucocorticoid used to counteract certain side effects of chemotherapies or as a direct chemotherapeutic agent in certain malignancies. It reduces lung tumor multiplicities in the tobacco smoke-exposed and non-exposed mice (*Witschi et al., 2005*). We showed that DEX inhibited CXCL13 production by epithelial cells and SPP1 production by TAMs (*Figure 3* and *Figure 6—figure supplement 1*), inhibited the EMT (*Figure 6—figure supplement 1*), reduced the tumor burden and prolonged the life span of the mice (*Figure 2*), revealing a new mechanism for this existing drug. Because DEX is a wide-spectrum anti-inflammatory drug that may cause severe side effects, more specific, CXCL13/CXCR5-targeting antibodies or small molecules should be developed to combat lung cancer, the leading cause of cancer-related mortality accounting for 1.59 million deaths worldwide in 2012 (*WHO, 2012*).

## Materials and methods

### Patient samples

The study was approved by the local research ethics committees of all participating sites; all lung cancer samples were collected with informed consent. A total of 201 previously untreated NSCLCs from HPRs or CRs were included (*Table 1*). The HPR patients were diagnosed in the last 5 years in participant hospitals in Yunnan Province, and the diagnosis of lung cancer was confirmed by at least three pathologists. Those who fulfilled the following criteria were selected for this study: (1) Residents of Xuanwei where the smoky coal was used. (2) Resided in their communities and never stayed in other regions for a long time (6 months or more). (3) Previously untreated primary lung cancer. (4) The tissue samples were taken at the time of surgery and quickly frozen in liquid nitrogen. The tumor samples contained a tumor cellularity of greater than 60% and the matched control samples had no tumor content. Serum samples were obtained from 80 untreated NSCLCs (40 from HPR and 40 from CR) and 40 healthy donors.

### Antibodies and reagents

The antibodies used in this work were: recombinant human CXCL13, human CXCL13 Quantikine ELISA Kit, mouse CXCL13 Quantikine ELISA Kit, mouse CXCL12 Quantikine ELISA Kit, mouse anti-human CXCR5-PE mAb, goat anti-human CXCL13 polyclonal Ab, goat anti-mouse CXCL13 polyclonal Ab (R&D, Minneapolis, MN), rat anti-mouse CD68-PE, rat anti-mouse CD45-APC, rat anti-mouse CD4-PerCP-CY5.5, rat anti-mouse CD19-FITC, rat anti-mouse CXCR5-PE/CY7 mAb (Biolegend, San Diego, CA), mouse anti-human CD68 mAb (Dako, Glostrup, Denmark), rabbit anti-mouse SPP1 polyclonal Ab (Proteintech, Chicago, IL), anti-TTF1 (Abcam, Cambridge, UK), rabbit anti-mouse E-Cadherin mAb, EMT Antibody Sampler Kit (Cell Signaling Technology, Beverly, MA), rabbit anti-human AhR polyclonal Ab, goat anti-human Lamin B polyclonal Ab, mouse anti-human α-tubulin mAb (Santa Cruz Biotechnology, Santa Cruz, CA), mouse anti-human β-actin antibody (Sigma, St. Louis, MO), Alexa Fluor 488 Donkey anti-Goat IgG (H+L), Alexa Fluor 555 Donkey Anti-Mouse IgG

(H+L), Alexa Fluor 647 Donkey anti-Rabbit IgG (H+L) (Life Technology, Thermo Fisher Scientific, Basingstoke, UK), and mouse SPP1 ELISA Kit (Shanghai GenePharma, Shanghai, China). BaP and DEX were purchased from Sigma.

## Assessment of CXCL13 expression

The expression of inflammatory factors in eight HPR patients (*Figure 1A*) was determined using the Human Inflammatory Cytokines & Receptors RT[2] Profiler PCR Array, which contains 84 cytokines/chemokines and their receptors. CXCL13 expression was detected in additional 193 HPR and CR patients, in BaP-treated cells, and in BaP-treated A/J, $Cxcl13^{-/-48}$ or $Cxcr5^{-/-50}$ mice by real-time PCR, Western blot analysis, ELISA, or IHC using anti-CXCL13 and anti-SPP1 antibodies. The immunoreactivity score (IRS) was calculated as IRS (0–12)=CP (0–4)×SI (0–3), where CP is the percentage of CXCL13-positive epithelial cells and SI is the staining intensity (*Remmele and Stegner, 1987*).

## Cell culture and preparation of conditioned medium

The lung adenocarcinoma cancer cell lines A549 and NCI-H1975 were obtained from ATCC (Manassas, VA, USA). Human normal bronchial epithelial cell line 16HBE was obtained from Clonetics (Walkersville, MD) and cultured according to standard protocols. The macrophages were obtained by incubation of THP-1 cells (ATCC) with 100 ng/mL phorbol 12-myristate 13-acetate (PMA) for 24 hr and then the culture medium was refreshed with serum-free Dulbecco's Modified Eagle Medium (DMEM) for another 24 hr (*Tsuchiya et al., 1982*). For preparation of conditioned medium, PMA-differentiated THP-1 cells were plated in six-well plates and co-cultured with CXCL13 for 48 hr. The medium was then centrifuged at 200×g for 10 min at 4°C to remove cell debris and stored at −80°C until use. For the analysis of EMT signaling by western blot analysis, supernatants were ultrafiltered for protein enrichment using a centrifugal filter device (Millipore, Darmstadt, Germany).

## Plasmids and transfections

We screened for transcription factors that could regulate *CXCL13* by website-based prediction (http://www.sabiosciences.com/chipqpcrsearch.php?species_id=0&factor=Over+200+TF&gene=CXCL13&nfactor=n&ninfo=n&ngene=n&B2=Search), and *CXCL13* promoter with wild-type or deletion mutation XRE-like element was cloned into pGL3-Basic vector. SPP1 was cloned into pcDNA3.1-flag vector. Cells were transfected with plasmids or siRNA using the Lipofectamine 2000 (Invitrogen, Frederick, MD) or lentivirus according to manufacturer's instruction. CXCL13 was cloned into the retroviral vector pGC-FU-GFP-IRES-Puromycin, and then transfected into the virus-packaging Phoenix cells. The supernatant was used to infect A549-Luc cells, and the infected cells were selected with 0.5 µg/mL Puromycin (Gene Oparation, Ann Arbor, MI) and 500 µg/mL G418 (Calbiochem, San Diego, CA).

## RT-PCR

The total RNA was isolated using the TRIZOL reagent (Invitrogen) and the phenol-chloroform extraction method according to the manufacturer's instruction. Total RNA (2 µg) was annealed with random primers at 65°C for 5 min. The cDNA was synthesized using a 1st-STRAND cDNA Synthesis Kit (Fermentas, Pittsburgh, PA). Quantitative real-time PCR was carried out using SYBR Premix ExTaq (Takara). The primer sequences for real-time PCR are listed in *Figure 1—source data 1*. Each sample was analyzed in triplicate three times.

## Migration

Trans-wells with 8-µm pores were incubated at 37°C in a $CO_2$ incubator for at least 1 hr. The differentiated THP-1 cells were seeded in the lower chamber, and A549-Luc-CXCL13 or H1975 cells ($1×10^4$) with serum-free DMEM/RPMI 1640 were added to the upper chamber and allowed to migrate for 24 hr. Cells on the inserts were fixed with 90% ethanol, stained with 0.0005% Gentian Violet Solution, and washed with PBS. Non-migrated cells on the upper side of the inserts were wiped off with a cotton swab. Migrated cells were counted in five microscopic fields at 4× magnification, and the counts were averaged (*Meijer et al., 2006*).

## Chromatin immunoprecipitation (ChIP) assay

A549 cells ($5\times10^6$) were seeded onto 10-cm dishes and treated with or without BaP (5 µM) for 2 h, and ChIP assay was performed as described (*Zhang et al., 2012*). Both input and immunoprecipitated DNA samples were analyzed by qPCR to determine the relative amounts of DNA from the *CXCL13* gene promoter region present in the samples. The primer pairs used here were listed in *Figure 1—source data 1*.

## Flow cytometry

Mouse lung cancer tissues were dissected into 2-mm fragments, followed by collagenase IV (0.2%, Sigma) digestion for 40 min at 37°C. A single-cell suspension was generated through a 200-mm-stainless steel wire mesh. The dissociated cancer cells labeled with indicated cell surface markers were sorted by MoFlo XDP Cell Sorter (Beckman Coulter, Brea, CA), and the data were analyzed on the Summit Software v5.0 (Beckman Coulter). All FACS analyses and sorting were paired with matched isotype control. Dead cells were excluded based on scatter profile.

## Immunofluorescence microscopy

Cells grown on coverslip (24 mm×24 mm) were fixed with 4% paraformaldehyde for 15 min, washed with 150 mM glycine in PBS, and permeabilized with 0.3% Triton X-100 in PBS for 20 min at room temperature. After blocking with 5% BSA, the cell smears were incubated with the indicated primary antibodies overnight at 4°C, washed, and FITC/PE-labeled secondary antibody in PBS was added to the cell smears. Images were taken by a laser scanning confocal microscopy (Zeiss, Oberkochen, Germany).

## Immunohistochemistry analysis

IHC assay was performed using anti-CXCL13 and anti-SPP1 antibodies as previously described (*Ma et al., 2011*). Briefly, formalin-fixed, paraffin-embedded human or mouse lung cancer tissue specimens (5 µm) were deparaffinized through xylene and graded alcohol, and subjected to a heat-induced epitope retrieval step in citrate buffer solution. The sections were then blocked with 5% BSA for 30 min and incubated with indicated antibodies at 4°C overnight, followed by incubation with secondary antibodies for 90 min at 37°C. Detection was achieved with 3, 3'-diaminobenzidine (DAB, Zhongshan Golden Bridge Biotechnology, Beijing, China) and counterstained with hematoxylin, dehydrated, cleared and mounted as in routine processing. The scoring of immunoreactivity was performed as described (*Remmele and Stegner, 1987*).

## ELISA

Concentration of CXCL13 in serum and cell culture supernatant was determined by ELISA using a commercially available ELISA kit (R&D). The absorbance of the plates was read at 450 nm using an automated microplate reader (Bio-Tek, Winooski, VT, USA).

## Microarray experiment and analysis

Eight pairs of human lung cancer tissues and their adjacent normal lung tissues were homogenized by Biopulverizer (Biospec, Bartlesville, OK). Total RNA was extracted from homogenized samples using the RNeasy Mini Kit (Qiagen, Valencia, CA). Total RNA (1 µg) was labeled and hybridized with a One-ColorQuick Amp Labeling Kit and Gene Expression Hybridization Kit (Agilent Technologies, Santa Clara, CA). Hybridization signals were detected using a DNA microarray scanner G2565BA (Agilent Technologies), and all scanned images were analyzed using Agilent Feature Extraction Software. Quantile normalization and subsequent data processing were performed using the GeneSpring GX v11.5.1 software package (Agilent Technologies). Quality assessment of mRNA data after filtering was carried out by Box Plot and Scatter-Plot. To identify differentially expressed mRNAs with statistical significance, Volcano Plot filtering between the two groups (fold change $\geq$ 1.5, *p* values $\leq$ 0.05) was performed. Pathway analysis was performed using the KEGG (Kyoto Encyclopedia of Genes and Genomes) database. The genes correlated with *CXCL13* were analyzed by Pearson's correlation coefficient, and the results were listed in *Figure 6—source data 1*.

## Western blotting

Cells were lysed on ice for 30 min in RIPA buffer (50 mM Tris-HCl pH 7.4, 150 mM NaCl, 0.1% SDS, 1% deoxycholate, 1% Triton X-100, 1 mM EDTA, 5 mM NaF, 1 mM sodium vanadate, and protease inhibitors cocktail), and protein extracts were quantitated. Proteins (20 µg) were subjected to 10–15% sodium dodecyl sulfate-polyacrylamide gel electrophoresis (SDS-PAGE), electrophoresed and transferred onto a nitrocellulose membrane. After blocking with 5% non-fat milk in Tris-buffered saline, the membrane was washed and incubated with the indicated primary and secondary antibodies and detected by Luminescent Image Analyzer LSA 4000 (GE, Fairfield, CO, USA).

## Animal studies

The animal studies were approved by the Institutional Review Board of Institute of Zoology, Chinese Academy of Sciences. All animal studies were conducted according to protocols approved by the Animal Ethics Committee of the Institute of Zoology, Chinese Academy of Sciences. Female NOD/SCID mice (5–6 weeks old) were purchased from Vital River Laboratory Animal Technology (Beijing, China). A/J mice (5–6 weeks old, female), homozygous Cxcl13-deficient, and homozygous Cxcr5-deficient mice were purchased from the Jackson Laboratory (Bar Harbor, Maine, USA). The exposure protocols were modified from previous studies (*Wattenberg and Estensen, 1996*; *Estensen and Wattenberg, 1993*). The A/J mice were exposed to BaP (0 mg/kg, 50 mg/kg, 100 mg/kg twice per week for 5 weeks) in corn oil via oral gavage, and treated with or without DEX (0.5 mg/kg per day for 10 weeks in diet; *Figure 2—figure supplement 1A*). The $Cxcl13^{-/-}$ and $Cxcr5^{-/-}$ mice were treated with BaP at 100 mg/kg twice per week for 8 weeks (*Figure 4—figure supplement 1B*). For microCT (PerkinElmer, Waltham, MA) and bioluminescence (using an IVIS Spectrum In Vivo Imaging System, PerkinElmer) analyses, mice were anesthetized by mixture of oxygen/isoflurane inhalation and positioned with legs fully extended, and assayed according to manufacturers' instruction. Survival of the mice was evaluated from the first day of treatment until death or became moribund, at which time points the mice were sacrificed.

## Statistical analysis

All statistical analyses were conducted using GraphPad Prism 5 (GraphPad Software, La Jolla, CA) and SPSS 12.0 software for Windows (Chicago, IL). Statistically significant differences were determined by Student's *t*-test, Wilcoxon rank sum test, one-way analysis of variance, the chi-squared test, or multivariate logistic analysis. *P* values less than 0.05 were considered statistically significant in all cases.

# Acknowledgements

The authors thank Dr. Fu Jia of PerkinElmer and Mrs. Qing Meng of our institute for their technical support. This work was supported by the National Natural Science Funds for Distinguished Young Scholar (81425025), the National Key Program for Basic Research (2012CB910800), the National Natural Science Foundation of China (81171925 and 81201537), and grants from the State Key Laboratory of Medical Genomics. The funders had no role in study design, data collection and analysis, decision to publish, or preparation of the manuscript.

# Additional information

### Funding

| Funder | Grant reference number | Author |
| --- | --- | --- |
| The National Natural Science Funds for Distinguished Young Scholar | 81425025 | Guang-Biao Zhou |
| The National Key Program for Basic Research | 2012CB910800 | Guang-Biao Zhou |
| National Natural Science Foundation of China | 81171925 | Guang-Biao Zhou |

| National Natural Science Foundation of China | 81201537 | Xin Cheng |
| Grants from the State Key Laboratory of Medical Genomics | | Guang-Biao Zhou |

The funders had no role in study design, data collection and interpretation, or the decision to submit the work for publication.

## Author contributions

G-ZW, XC, BZ, Z-SW, Y-CH, H-BC, G-FL, Z-LH, Y-CZ, LF, M-MW, L-WQ, YC, Acquisition of data; G-BZ, Conception and design, Analysis and interpretation of data, Drafting and revising the article

## Ethics

Human subjects: The study was approved by the local research ethics committees of all participating sites; all lung cancer samples were collected with informed consent.

Animal experimentation: Animal studies were conducted according to protocols approved by the Animal Ethics Committee of the Institute of Zoology, Chinese Academy of Sciences, with the approval ID of AEC2010070202.

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
