## [Decision Letter]

Thank you for submitting your work entitled "The chemokine CXCL13 in lung cancers associated with environmental polycyclic aromatic hydrocarbons pollution" for peer review at *eLife*. Your submission has been favorably evaluated by Sean Morrison (Senior editor) and three reviewers, one of whom served as Guest Reviewing editor.

The reviewers have discussed the reviews with one another and the Reviewing editor has drafted this decision to help you prepare a revised submission.

Summary:

All three reviewers felt the work was interesting and of importance, but had concerns about several aspects of the paper that need to be addressed before considering publication. In particular, all three reviewers asked for more detailed analysis of *Cxcl13* and *Cxcr5* mouse knockout studies, an essential component of the paper. In addition, analyses of *CXCL13* levels in orthogonal datasets were requested for Figure 1, and additional controls were recommended for ChIP data in Figure 3. Another issue raised was the source of *CXCL13*, whether it was derived from epithelial cells or associated immune cells.

Essential revisions:

1) Figure 1 – is *CXCL13* enriched in orthogonal datasets or is this specific to the Chinese cohorts? We would suggest analysis of TCGA data of tumor versus normal tissue and smokers versus non-smokers. It would also be of interest to perform a similar comparison in microarray data deposited from genetically engineered mouse models of lung cancer to see whether in these models, which are not carcinogen induced, if *CXCL13* is not enriched. For example, mouse GEO datasets GSE6135, GSE21581, GSE54353 are from lung tumors derived following LKB1 inactivation. It would be interesting to know whether or not *CXCL13* is involved in this context.

2) What are the expression levels of *CXCR5* in human and mouse samples with lung cancers?

3) It is unclear which cell type is staining positive for *CXCL13* in Figure 2. It would be useful to add IHC panels to help determine whether it is being produced from tumor cells or tumor associated macrophages. The authors could use TTF1 stain to mark lung cancer cells, and CD68 stain to mark macrophages. Alternatively they could perform flow cytometry and sort by EPCAM or CD68.

4) Figure 3 needs additional controls. The authors should use qPCR instead of PCR to present the ChIP data. In addition, although the specific induction of binding by BAP is nice, the authors also need to test another gene without the XRE element, perhaps *CXCL12,* which was used in another experiment. For the EMSA, it seems that the lane with the shift is overloaded. This blot is not convincing and needs to be repeated. Alternatively, if the results are not robust, they should be removed from the paper, since it erodes confidence in the ChIP result.

5) Figure 4 which is a key component of this paper should include quantification of tumor burden by microCT (as presented in Figure 2—figure supplement 1). In the CT images presented it appears there are tumors in *Cxcl13^-/-^* mice. Also at what time point were CTs performed? There should also be more rigorous quantification of tumor nodules in histologic sections between mouse groups (e.g. see Tan et al. PLOS One 2013; PMID 24260500). This data needs to be presented for the *Cxcr5* mice as well. Also for Figure 3—figure supplement 1 that is used to document *Cxcl13* inactivation; which band is it? If it is the lower band why is it higher in the hetereozygous mice? Same for *Cxcr5* mice.

6) As in point 2, additional IHC controls for Figure 6 would help to confirm that macrophages are indeed the cell type positive for SPP1 in the BaP induced tumors. As above, the authors could use TTF1 stain to mark lung cancer cells and CD68 stain to mark macrophages, or perform flow cytometry and sort by EPCAM or CD68.

---

## [Author Response]

*Summary:*

*All three reviewers felt the work was interesting and of importance, but had concerns about several aspects of the paper that need to be addressed before considering publication. In particular, all three reviewers asked for more detailed analysis of* cxcl13 *and* cxcr5 *mouse knockout studies, an essential component of the paper. In addition, analyses of* CXCL13 *levels in orthogonal datasets were requested for Figure 1, and additional controls were recommended for ChIP data in Figure 3. Another issue raised was the source of* CXCL13*, whether it was derived from epithelial cells or associated immune cells.*

We performed additional and more detailed analysis in *Cxcl13* and *Cxcr5* knockout mice, analyzed *CXCL13* levels in orthogonal datasets, investigated the source of CXCL13, and used additional controls for ChIP data. In addition to the above revisions, 14 new references were added to the paper, which now has 3 tables, 6 figures, 77 references, 2 supplementary tables and 5 supplementary figures.

*Essential revisions:*

*1) Figure 1 – is* CXCL13 *enriched in orthogonal datasets or is this specific to the Chinese cohorts? We would suggest analysis of TCGA data of tumor versus normal tissue and smokers versus non-smokers. It would also be of interest to perform a similar comparison in microarray data deposited from genetically engineered mouse models of lung cancer to see whether in these models, which are not carcinogen induced, if* CXCL13 *is not enriched. For example, mouse GEO datasets GSE6135, GSE21581, GSE54353 are from lung tumors derived following LKB1 inactivation. It would be interesting to know whether or not* CXCL13 *is involved in this context.*

We thank the reviewers for the important suggestions, and evaluated the expression of *CXCL13* in tumor versus normal tissue and smokers versus non-smokers in TCGA data and a cancer microarray database Oncomine (www.oncomine.org). In two works of TCGA (Nature 2014; Nature 2012), *CXCL13* expression was assessed by microarray or RNA-seq, and the results showed that *CXCL13* was not up-regulated in tumor samples compared to normal tissues (data not shown). Nevertheless, *CXCL13* in tumor samples of smokers was not increased compared to that in nonsmokers (data not shown). However, in several works documented in the Oncomine database, *CXCL13* was elevated in tumor compared to normal tissues (Figure 1), and was increased in smokers compared to non-smokers (Figure 1). These results suggested that overexpression of *CXCL13* was not specific to the Chinese cohorts.

In mouse GEO dataset GSE54353, *Cxcl13* expression in wild type (normal), *Kras*^G12D^, and *Lkb1(Stk11*)^-/-^*Pten*^-/-^ mice was not significantly different (Figure 1). In GSE6135, *Cxcl13* in *Kras*^G12D^*Stk11*^L/+, -/-, L/-^ mice was slightly but not statistically significantly higher than in *Kras*^G12D^*Stk11*^wt^ mice (p=0.08; Figure 1). In GSE21581 dataset, *Cxcl13* was slightly elevated in primary tumors and significantly increased in metastatic tumors of the *Kras*^G12D^*Stk11*^-/-^ mice, as compared to that of the *Kras*^G12D^ mice (Figure 1). These results suggested that *Cxcl13* might play a role in *Stk11*-related lung carcinogenesis which warrants further investigation.

*2) What are the expression levels of* CXCR5 *in human and mouse samples with lung cancers?*

We tested the expression of *CXCR5* in 24 NSCLCs by qPCR, and found that in tumor samples it was slightly higher than in paired normal lung tissues (Figure 4). In tumor samples of mice treated with BaP at 50 or 100 mg/kg, *Cxcr5* was also slightly up-regulated (Figure 4).

*3) It is unclear which cell type is staining positive for CXCL13 in Figure 2. It would be useful to add IHC panels to help determine whether it is being produced from tumor cells or tumor associated macrophages. The authors could use TTF1 stain to mark lung cancer cells, and CD68 stain to mark macrophages. Alternatively they could perform flow cytometry and sort by EPCAM or CD68.*

We thank the reviewers for the comments and performed IHC and immunofluorescence assays to determine the source of Cxcl13. By IHC assay, we found that both the Cd68 positive macrophages and Ttf1 positive lung cancer cells were stained positive for Cxcl13, but Ttf1 positive cells constituted more than 95% of the cellular component of the lung tumor tissues of the mice (Figure 2). This observation was confirmed by immunofluorescence assay (Figure 2) using antibodies against Cxcl13 (green), Cd68 (red), and Ttf1 (white). DAPI was used to counter stain the nucleus. These results indicate that Ttf1 positive lung cancer cells were the main source of Cxcl13 in mice exposed to BaP.

*4) Figure 3 needs additional controls. The authors should use qPCR instead of PCR to present the ChIP data. In addition, although the specific induction of binding by BAP is nice, the authors also need to test another gene without the XRE element, perhaps* CXCL12*, which was used in another experiment. For the EMSA, it seems that the lane with the shift is overloaded. This blot is not convincing and needs to be repeated. Alternatively, if the results are not robust, they should be removed from the paper, since it erodes confidence in the ChIP result.*

We thank the reviewers for the comments, and performed qPCR to test the expression of *CXCL13, CXCL12, CXCL14, CXCL11* and *CXCL2* in ChIP experiment of cells co-incubated with or without BaP. The results showed that among these chemokines, only *CXCL13* was enriched and significantly up-regulated by BaP treatment (Figure 5 here and Figure 3 of the paper). We used this data to replace the PCR results of the original Figure 3. The EMSA result was removed from the paper.

*5) Figure 4 which is a key component of this paper should include quantification of tumor burden by microCT (as presented in Figure 2—figure supplement 1). In the CT images presented it appears there are tumors in* cxcl13^-/-^
*mice. Also at what time point were CTs performed? There should also be more rigorous quantification of tumor nodules in histologic sections between mouse groups (e.g. see Tan et al. PLOS One 2013; PMID 24260500). This data needs to be presented for the* cxcr5 *mice as well. Also for Figure 3–figure supplement 1 that is used to document* cxcl13 *inactivation; which band is it? If it is the lower band why is it higher in the hetereozygous mice? Same for* cxcr5 *mice.*

We thank the reviewers for the important comments and suggestions, and evaluated the tumor burden of the *Cxcl13*- and *Cxcr5*-deficiency mice. Firstly, tumor nodules in histologic sections of mice upon BaP treatment were analyzed as describe (Tan et al. PLOS One 2013), and the result showed that at treatment time points of 120, 180, and 240 days, *Cxcl13*^-/-^ mice had much less lesions than *Cxcl13*^+/-^ and *Cxcl13*^+/+^ mice (Figure 4). For microCT analysis of the mice treated with BaP for 240 days, the tumor volume of *Cxcl13*^-/-^ mice was significantly lower than *Cxcl13*^+/-^ and *Cxcl13*^+/+^ mice (Figure 4). Tumor burden of *Cxcr5*^-/-^ mice, reflected by tumor nodules and microCT assays, was also significantly lower than *Cxcr5*^+/-^ and *Cxcr5*^+/+^ mice (Figure 4).

The *Cxcl13*^-/-^ mice were obtained from the Jackson Laboratory (https://www.jax.org/strain/005626). The mice were also called B6.129X1-*Cxcl13^tm1Cys^*/J. A targeting vector was constructed in which base pairs 18-116 of exon 2 from the endogenous gene were replaced with an in-frame stop codon, a Mengo virus internal ribosome entry site, an enhanced green fluorescent protein gene (EGFP), and a loxP-flanked neomycin resistance gene. The construct was electroporated into 129X1/SvJ derived JM-1 embryonic stem (ES) cells. Correctly targeted ES cells were injected into C57BL/6J blastocysts and the resulting chimeric males were backcrossed for germ-line transmission to C57BL/6J females. Offspring were mated with Cre-expressing C57BL/6J to remove the neomycin resistance gene. The neo-excised heterozygotes were backcrossed to C57BL/6J for ten generations before being made homozygous. Mice that are homozygous for the targeted mutation are viable, fertile and normal in size. No endogenous gene product (mRNA or protein) is detected, EGFP is not expressed. The primers to detect *Cxcl13* were designed by the Jackson Laboratory, and the expected results were: mutant=~240 bp, heterozygote =~240 bp and ~200 bp, wild type=~200 bp (https://www2.jax.org/protocolsdb/f?p=116:2:::NO:2:P2_MASTER_PROTOCOL_ID,P2_JRS_CODE:945,005626). Our results were in consistent with the references provided by the Jackson Laboratory, e.g., the higher one of Figure 4—figure supplement 1 was the inactivation band.

For *Cxcr5^-/-^* mice (https://www.jax.org/strain/006659), a targeting vector was designed to replace the 350 bp coding region of exon 2 of the targeted gene with a neomycin resistance gene. The construct was electroporated into 129S2/SvPas-derived D3 embryonic stem (ES) cells. Correctly targeted ES cells were aggregated with morulae from outbred CD-1 mice and then transferred into pseudopregnant CD-1 females. Chimeric mice were bred with CD-1. The donating investigator reported that mutant mice were backcrossed to C57BL/6 mice for 8 generations prior to arrival at The Jackson Laboratory. Homozygous (CXCR5-deficient) mice are viable and fertile. No endogenous gene product (mRNA or protein) is detected. In Figure 4—figure supplement 1, our results were in consistent with the references provided by the Jackson Laboratory, e.g., the lower one was the inactivation band.

*6) As in point 2, additional IHC controls for Figure 6 would help to confirm that macrophages are indeed the cell type positive for SPP1 in the BaP induced tumors. As above, the authors could use TTF1 stain to mark lung cancer cells and CD68 stain to mark macrophages, or perform flow cytometry and sort by EPCAM or CD68.*

We thank the reviewers for the comments and performed IHC and immunofluorescence assays to determine the source of Spp1. By IHC assay, we found that the Cd68 positive macrophages were strongly stained by Spp1, whereas Ttf positive lung cancer cells were weakly stained by Spp1 (Figure 6). This observation was confirmed by immunofluorescence assay (Figure 7B) using antibodies against Spp1 (green), Cd68 (red), and Ttf (white). DAPI was used to counter stain the nucleus. These results indicate that Cd68 positive macrophages were the main source of Spp1 in mice exposed to BaP.